# Hepatorenal Tyrosinaemia: Impact of a Simplified Diet on Metabolic Control and Clinical Outcome

**DOI:** 10.3390/nu13010134

**Published:** 2020-12-31

**Authors:** Friederike Bärhold, Uta Meyer, Anne-Kathrin Neugebauer, Eva Maria Thimm, Dinah Lier, Stefanie Rosenbaum-Fabian, Ulrike Och, Anna Fekete, Dorothea Möslinger, Carmen Rohde, Skadi Beblo, Michel Hochuli, Nina Bogovic, Vanessa Korpel, Stephan vom Dahl, Sebene Mayorandan, Aleksandra Fischer, Peter Freisinger, Katharina Dokoupil, Margret Heddrich-Ellerbrok, Monika Jörg-Streller, Agnes van Teeffelen-Heithoff, Janina Lahl, Anibh Martin Das

**Affiliations:** 1Department of Paediatrics, Hannover Medical School, Carl-Neuberg-Straße 1, 30625 Hannover, Germany; friederike.baerhold@gmx.de (F.B.); meyer.uta@mh-hannover.de (U.M.); sebene.mayorandan@ukmuenster.de (S.M.); 2Klinik für Allgemeine Pädiatrie, Universitätsklinikum Düsseldorf, Moorenstraße 5, 40225 Düsseldorf, Germany; neugebauer@med.uni-duesseldorf.de (A.-K.N.); eva.thimm@med.uni-duesseldorf.de (E.M.T.); 3Klinik für Kinder- und Jugendmedizin, Stoffwechselzentrum, Klinikum am Steinenberg, Steinenbergstraße 31, 72764 Reutlingen, Germany; lier_d@klin-rt.de (D.L.); fischer_al@klin-rt.de (A.F.); Freisinger_P@klin-rt.de (P.F.); 4Zentrum für Kinder- u. Jugendmedizin, Universitätsklinikum Freiburg, Mathildenstraße 1, 79106 Freiburg, Germany; stefanie.rosenbaum-fabian@uniklinik-freiburg.de; 5Klinik für Kinder- und Jugendmedizin, Universitätsklinikum Münster, Albert-Schweitzer-Campus 1, 48149 Münster, Germany; ulrike.och@ukmuenster.de (U.O.); vanteeff@uni-muenster.de (A.v.T.-H.); 6Kinder- und Jugendheilkunde, AKH Universitätsklinikum Wien, Währinger Gürtel 18-20, 1090 Wien, Austria; anna.fekete@meduniwien.ac.at (A.F.); dorothea.moeslinger@meduniwien.ac.at (D.M.); 7Universitätsklinik für Kinder und Jugendliche, Universitätsklinikum Leipzig, Liebigstraße 20 a, 04103 Leipzig, Germany; carmen.rohde@medizin.uni-leipzig.de (C.R.); skadi.beblo@medizin.uni-leipzig.de (S.B.); 8Klinik für Endokrinologie, Diabetologie und Klinische Ernährung, Universitätsspital Zürich, Rämistraße 100, 8091 Zürich, Switzerland; michel.hochuli@insel.ch; 9Department of Diabetes, Endocrinology, Nutritional Medicine and Metabolism Inselspital Bern, University Hospital and University of Bern, Freiburgstrasse 15, 3010 Bern, Switzerland; 10Klinik für Gastroenterologie, Hepatologie und Infektiologie, Universitätsklinikum Düsseldorf, Moorenstraße 5, 40225 Düsseldorf, Germany; nina.bogovic@med.uni-duesseldorf.de (N.B.); vanessa.korpel@med.uni-duesseldorf.de (V.K.); Stephan.Dahl@med.uni-duesseldorf.de (S.v.D.); 11Dr. von Haunersches Kinderspital, Lindwurmstraße 4, 80337 München, Germany; katharina.dokoupil@med.uni-muenchen.de; 12Klinik für Kinder- und Jugendmedizin, Universitätsklinikum Hamburg Eppendorf, Martinistraße 52, 20246 Hamburg, Germany; heddrich@uke.uni-hamburg.de; 13Department für Kinder- und Jugendheilkunde, Medizinische Universität Innsbruck, Anichstraße 35, 6020 Innsbruck, Austria; monika.joerg-streller@tirol-kliniken.at; 14Nutricia GmbH, Metabolics Expert Centre^D-A-CH^, Am Hauptbahnhof 18, 60329 Frankfurt, Germany; janina.lahl@danone.com

**Keywords:** hepatorenal tyrosinaemia, tyrosinaemia type 1, low protein diet, phenylalanine, tyrosine, inborn error of metabolism, compliance

## Abstract

*Background*: Tyrosinaemia type 1 is a rare inherited metabolic disease caused by an enzyme defect in the tyrosine degradation pathway. It is treated using nitisinone and a low-protein diet. In a workshop in 2013, a group of nutritional specialists from Germany, Switzerland and Austria agreed to advocate a simplified low-protein diet and to allow more natural protein intake in patients with tyrosinaemia type 1. This retrospective study evaluates the recommendations made at different treatment centers and their impact on clinical symptoms and metabolic control. *Methods*: For this multicenter study, questionnaires were sent to nine participating treatment centers to collect data on the general therapeutic approach and data of 47 individual patients treated by those centers. *Results*: Dietary simplification allocating food to 3 categories led to increased tyrosine and phenylalanine blood concentrations without weighing food. Phenylalanine levels were significantly higher in comparison to a strict dietary regimen whereas tyrosine levels in plasma did not change. Non-inferiority was shown for the simplification and liberalization of the diet. Compliance with dietary recommendations was higher using the simplified diet in comparison to the stricter approach. Age correlates negatively with compliance. *Conclusions*: Simplification of the diet with increased natural protein intake based on three categories of food may be implemented in the diet of patients with tyrosinaemia type 1 without significantly altering metabolic control. Patient compliance is strongly influencing tyrosine blood concentrations. A subsequent prospective study with a larger sample size is necessary to get a better insight into the effect of dietary recommendations on metabolic control.

## 1. Introduction

Tyrosinaemia type 1 (Tyr 1) is a rare inherited metabolic disease. With an estimated incidence of 1:120,000 in Europe it belongs to the group of orphan diseases [1]. Tyr 1 is caused by an enzyme defect of fumarylacetoacetate hydrolase (FAH) in the tyrosine degradation pathway. The defect of FAH leads to the accumulation of tyrosine and its toxic metabolites such as succinylacetone (SA). SA serves as a surrogate parameter of toxicity. Toxic metabolites cause organ damage especially to the kidneys, liver and nervous system [1,2]. Affected patients show an increased incidence of impaired cognitive function and compromised intelligence [3,4,5]. The therapy of Tyr 1 consists of nitisinone intake (2-(2-nitro-4-trifluormethylbenzoyl)-1,3-cyclohexanedione, NTBC), low-protein diet and a supplemental amino acid mixture [6,7,8]. Nitisinone is an inhibitor of the tyrosine degradation pathway and therefore leads to an increase of tyrosine blood concentrations [9]. Phenylalanine may be supplemented if needed. Van Ginkel et al. suggested that high phenylalanine concentrations might protect from neurocognitive impairments, higher concentrations than generally assumed may be needed to normalize cerebral phenylalanine levels [10].

Based on the lethal natural course of the disease physicians tended to be very strict regarding dietary recommendations. Calculation of dietary tyrosine uptake was often advised. Several clinical recommendations were published (e.g., De Laet et al. (2013), Mayorandan et al. (2014), Chinsky et al. (2017)) in an attempt to standardize the treatment of Tyr 1. However, different approaches are still followed regarding dietary therapy and monitoring of Tyr 1 at the different metabolic centers due to a lack of scientifically based guidelines [11]. In some centers, a very restrictive diet is recommended, whereas in other centers food with a higher protein content is allowed. Furthermore, it is unclear, whether a single daily dose of nitisinone is sufficient and what plasma levels of nitisinone are required to suppress toxic metabolites [12,13]. The aim of our study was to assess the impact of a liberalized and simplified diet on metabolic control.

## 2. Materials and Methods

In 2013, data from 42 patients were analyzed and discussed during a workshop by pediatricians and nutritionists from pediatric units in Germany (n = 8), Switzerland (n = 1) and Austria (n = 2) (see Appendix A). Participants had identified a major problem with dietary compliance in Tyr 1 patients, particularly those prescribed a strict diet. In 2013, a considerable number of 17 out of 42 patients/families (40%) were weighing foods to calculate protein intake (see Appendix A). Many patients did not consume low-protein food (40%) but inappropriate items like regular pastries and pasta (43%), milk and nuts (Figure A2 and Figure A3). Compliance was much better in younger children whose diet was controlled by the parents than in older children over 7 years of age (Figure A4).

The workshop’s aim was to improve dietary compliance by encouraging families and patients not to calculate tyrosine, phenylalanine or protein intake, not to weigh foods and allow moderate protein foods according to appetite. Food was divided into three groups according to protein content: low-protein food (unrestricted), moderate-protein food (occasionally appropriate) and protein-rich food (inappropriate). There was consensus that meat, sausages and eggs were inappropriate, whereas fruits, vegetables, potatoes, rice, butter, oil, cream, crème fraiche, sour cream, low-protein food (e.g., bread, pasta, pastries, milk), rice and oat drinks were regarded as appropriate. There was no general consensus on normal milk, yoghurt, cheese, normal bread, pasta and pastries, legumes, nuts and seeds, so it was agreed that these may be taken occasionally, i.e., not more than twice a week in low to moderate amounts (Table 1). More moderate-protein food could be allowed on an individual basis if blood levels of tyrosine were in the therapeutic range (<400 µmol). It was assumed, that these relaxed recommendations result in higher dietary compliance and quality of life without hampering metabolic control.

So far, it has not been assessed which metabolic centers in Switzerland, Austria and Germany (DACH region) have changed their recommendations in response to the workshop and how these changes affected metabolic control. The present survey compared different treatment strategies and evaluated their impact on the patient’s clinical condition and metabolic control in terms of tyrosine and phenylalanine concentrations in blood. The aim was to verify the workshop’s hypothesis that not calculating natural protein intake and a relaxed, simplified diet results in increased compliance and is not inferior to a strict diet in terms of metabolic control.

The participating centers were asked to complete questionnaires for the 2011–2013 (prior to the workshop) and 2016–2019 periods (post-workshop) including the following items: (1) general dietary recommendations in the center, (2) long term tyrosine and phenylalanine blood concentrations in individual patients, (3) nitisinone dose and compliance to medication and (4) individual patient compliance in terms of dietary restrictions (simplified versus strict) and medication intake.

The primary outcome parameters were mean blood concentrations of phenylalanine and tyrosine taken every 2–3 months in younger patients and about every 3–6 months in older patients. In some older patients with good metabolic control, blood samples were only taken once per year. The influence of the dietary recommendations on the outcome measures was analyzed using linear regression. Other possible confounding variables such as nitisinone intake, compliance, date of birth and age at diagnosis were analyzed separately. A *t*-test for dependent samples was performed if patient data were available for both 2011–2013 and 2016–2019 in order to analyze the impact of altered recommendations on the outcome measures.

The maximum and minimum tyrosine and phenylalanine concentrations in the respective time periods were recorded. Data were retrospectively collected by treating physicians and nutritionists from medical records. Compliance was assessed retrospectively by dieticians, nutritionists or the treating physician using a 10-point Likert scale. Patients were grouped by their type of dietary recommendations (strict or simplified). The significance level was set at alpha = 5%.

A positive ethical vote was obtained from the ethical review board of Hanover Medical School as the master commission (EC no. 8264_BO_K_2019). Ethical votes were sought from local ethical bodies of the other participating centers as required.

## 3. Results

Nine centers from Germany, Austria and Switzerland completed the questionnaires for a total number of 47 patients diagnosed with Tyr 1. Each center included an average of 5.2 patients (min: 2, max: 10 patients). 64% of the patients (n = 30) were male, 36% (n = 17) female; mean year of birth: 2005 (min: 1988, max: 2018, std.-dev.: 7.4). The mean age at diagnosis was 5.7 months (min: 0, max: 36 months, std.-dev.: 7.7). Three of the patients received liver transplantation, so they were excluded from the evaluation of tyrosine and phenylalanine blood concentrations. The following sections give an overview of the most important findings.

### 3.1. Dietary Recommendations and Target Parameters

The target values for tyrosine ranged from <400 to <800 µmol/L. Six of the centers recommended tyrosine levels of <400, two <500 µmol/L and one center aimed for values <800 µmol/L, phenylalanine values should be in the reference range (30–240 μmol/L, depending on age and laboratory). These recommendations have not changed compared to the period before 2013. All centers recommended a division of the amino acid mixture into three doses per day.

The diet in an individual center was simplified when foods were classified in three categories: unrestricted, occasionally appropriate and inappropriate. All participating centers were asked to categorize the following 21 food items: vegetables, fruits, cream, yoghurt/milk, cream cheese, regular pastries, potato/rice, butter/oil/mayonnaise, egg in food, meat/sausage, cheese, fish/seafood, nuts, rice drink/oat drink, egg, poultry, low protein special foods, low protein milk, sugar-rich drinks, food with <2 g protein/100 g and food with 2–6 g protein/100 g. The more foods were categorized as unrestricted, the more relaxed the diet.

Only four of the nine participating centers changed their food recommendation after the workshop in 2013, three markedly relaxed them, one center did no longer recommend protein calculation. Protein in meals was either estimated or food was weighed and protein content calculated by patients and families. None of the centers recommended calculation of phenylalanine in meals. Table 2 gives an overview of protein calculation, phenylalanine and tyrosine target values and diet for each center.

Center 9 clearly stands out from the others as meat, sausage, poultry, fish and cheese were classified as occasionally appropriate while the other centers classified these as inappropriate. Interestingly, Center 9 tolerated tyrosine levels up to 800 µmol/L, much higher than in the other centers, both patients treated by Center 9 had tyrosine levels in their respective therapeutic range.

From 2016–2019 centers 2, 6 and 8 achieved the lowest mean tyrosine blood concentrations (447, 476 and 464 µmol/L, respectively). Based on these low levels, centers 6 and 8, relaxed the diet by allowing regular pastries. This shows that the recommendations given in Table 1 can be relaxed on an individual basis without losing metabolic control.

### 3.2. Medical Formula Intake and Compliance

Between 2011 and 2013 the mean daily dose of NTBC was 0.78 mg/kg body weight per day. It increased to 0.84 mg/kg for the 2016–2019 period. Patients receiving strict dietary recommendations during this period also took the largest amount of NTBC (0.92 mg/kg bodyweight per day). Patients who were treated by centers which switched to a simplified diet took the smallest amount (0.63 mg/kg bodyweight per day). The maximum daily dose prescribed was 1.8, the minimum was 0.2 mg/kg per day. 23 patients took a single dose (mean year of birth: 2005), 21 patients two doses (mean year of birth: 2006) and only two patients three doses per day (mean year of birth: 2003). The number of doses per day varied within centers.

Protein intake via amino acid mixture was recommended by dieticians according to DACH 2015 (33%) or DGE 1985 (44%). All centers suggested to divide amino acid mixture into three doses per day.

Compliance for each patient was judged by the treating physician or nutritionist for diet and medication intake on a 10-point Likert scale, respectively. Compliance with medication was significantly higher (average of 9.0 points) compared to dietary compliance (6.5 points) in the 2016–2019 period. There were considerable differences between centers ranging from 4 points to 10 points. The dietary compliance was at all times higher in the group receiving simplified dietary recommendations (diet 2011–2013: 7.9 vs. 5.9 points with strict diet, 2016–2019: 6.9 vs. 6.1 points with strict diet). Using linear regression, age was found to have a significant impact on compliance (r = 0.48; *p* = 0.001).

### 3.3. Patient Data

#### Laboratory Findings

After the newborn period, patients below the age of 12 months were seen twice a year by all centers for blood collection and dietary counseling. One center reduced these appointments to one per year for patients > 1 year, two more centers for patients > 7 years. 5 centers controlled their patients twice a year even above the age of 18 years. The mean tyrosine and phenylalanine blood concentrations are shown in Table 3 and Table 4. Mean plasma tyrosine concentrations were above the target values of the respective center (Table 2).

In the 2016–2019 period mean phenylalanine concentrations were significantly higher in the group treated with a simplified diet (std. dev.: 9.1; CI: [2.35–39.08]; *p*: 0.028) indicating that the amount of natural protein was increased compared to a strict diet (Table 4). The mean tyrosine blood concentrations in the period 2016–2019 did not differ significantly between the groups with a strict or simplified/relaxed diet (std. dev.: 46.32; CI: [−93.17–93.79]; *p* = 0.995) (Table 4).

Regarding separately the group of patients that switched to a simplified relaxed diet after the workshop (n = 9) showed both increased tyrosine and phenylalanine levels, indicating a higher intake of natural protein. Only the difference in tyrosine comparing both periods, 2011–2013 and 2016–2019, was statistically significant. The tyrosine levels rose from 432 µmol/L average in 2011–2013 to 580 µmol/L in 2016–2019 (mean: 121; std. dev.: 101.5; CI: [43.87–199.9]; *p* = 0.007) (Figure 1a). However, metabolic control under a relaxed diet was not inferior to strict diet in the 2016–2019 period (Table 4). Mean phenylalanine levels increased from 48 µmol/L to 69 µmol/L (mean: 26.9; std. dev.: 51.1; *p* = 0.153) (Figure 1b).

Patients who constantly received strict dietary recommendations (n = 14) and those who constantly received simplified dietary recommendations (n = 6) during the whole observation period did not show significant changes of tyrosine and phenylalanine blood concentrations, compliance or nitisinone intake.

Succinylacetone was detected in blood samples of six patients in the 2011–2013 period and in nine patients in the 2016–2019 period. This means, that SA was detected in 33.3% of patients receiving strict recommendations and 5% of patients receiving simplified recommendations in 2016–2019.

Neurocognitive impairment was observed in 10 patients though no formal testing was done. This clinical feature has previously been linked to elevated tyrosine and reduced phenylalanine levels [4,10]. In our study, there was no correlation of high tyrosine and low phenylalanine levels with poor neurocognitive outcome as judged by the health care providers.

## 4. Discussion

Non-inferiority was shown for dietary treatment of Tyr 1-patients with a simplified diet based on three categories of food. There were no significant differences between mean tyrosine concentrations in 2016–2019 for strict (mean: 5551 µmol/L) and relaxed diet (mean: 552 µmol/L). For the centers that did change from a strict to a more relaxed diet there was a significant increase in tyrosine concentrations but to a level of 580 µmol/L in line with both the strict and relaxed diets. Strict dietary recommendations did not lead to significantly lower tyrosine levels compared to the group receiving simplified recommendations. This could be due to lower compliance in the group under strict diet. The group receiving strict dietary recommendations were prescribed the highest average dose of NTBC (0.92 mg/kg body weight per day in comparison to 0.63 mg/kg body weight per day in the group receiving simplified recommendations).

Not all centers which participated in the workshop in 2013 switched from a strict diet to a relaxed diet. The treating physicians and nutritionists may have been reluctant to change the diet in an attempt to maintain optimal metabolic control. In the period 2011–2013 tyrosine levels were even lower under a simplified diet (Figure 1a). It is conceivable that only patients with low mean tyrosine plasma levels chose to relax their dietary recommendations after the workshop in 2013. This may explain the rise in tyrosine levels after the switch.

Overall, a simplification of the dietary recommendations was only implemented by five centers. Centers sticking to a stricter diet also used higher amounts of nitisinone which may reflect a more cautious approach in these centers. In addition to the medical aspects, the quality of life should also be considered. A greater choice of food facilitates social participation. Furthermore, fewer special low-protein foods have to be purchased, which is a financial relief.

Phenylalanine levels in the 2016–2019 period were higher under a simplified diet, showing again non-inferiority of a relaxed diet. Increased phenylalanine levels are thought to be protective in terms of neurocognitive function [10]. After the change of diet, phenylalanine blood concentrations were predominantly within the normal range.

A higher compliance to diet was found in the group receiving a simplified/relaxed diet compared to patients on a strict diet in the 2011–2013 period. In the 2016–2019 period the compliance decreased in the group of patients that switched to a relaxed diet after 2013. We expected the compliance to increase after relaxation of the diet. Since age is a confounding factor for compliance this might be one explanation for this surprising result. On the other hand, compliance was subjectively judged by healthcare professionals who may interpret simplified diet as poor compliance. As could be expected, pharmacological compliance was better compared to dietary compliance. A strong correlation was found between patients’ adherence to diet and tyrosine blood concentrations. Already in the workshop in 2013, poor dietary compliance was identified by the participating dieticians and nutritionists as a major unmet need in the treatment of tyrosinaemia patients. Therefore, the main focus should be on increasing compliance to achieve improved metabolic control rather than further restricting food choices. Protein tolerance has been shown to increase with age which argues in favor of a simplified and relaxed diet especially in adolescents and adults [14].

There were some limitations in our study. Since the information on dietary compliance was collected by a questionnaire, it is possible that the results are biased. It can be assumed, that the medical personnel answering the questionnaire did not apply the 10-point Likert scale, used to asses compliance, uniformly. Compliance might have been interpreted differently by the centers. Data were collected for the last 9 years, therefore not all details may have been recalled correctly. Another difficulty was caused by the frequency of clinical appointments. Older patients were seen less frequently than younger patients (once or twice per year). The laboratory findings recorded at the outpatient visit do not necessarily reflect the situation of the last 6–12 months at home. It was not recorded how much amino acid mixture the patients took per kg bodyweight. Accordingly, this aspect was not considered in the evaluation, but could have a relevant influence on metabolic control and outcome. Another confounding factor was age. In this retrospective study, dietary assessments were not performed at standardized intervals. Therefore, the exact amount of appropriate, inappropriate and occasionally appropriate foods could not be quantified.

The study population of 44 patients included in the calculations is relatively large for a study on an ultrarare orphan disease. Still, the small sample size with at times only two patients per metabolic center and the retrospective character of this survey are clear limitations of our study. In the future, prospective studies in a larger group of hepatorenal tyrosinaemia patients should be performed including quality of life issues. The present survey was neither controlled nor randomized. In future studies, a waiting-control group could be implemented following an unaltered diet before being randomized to strict and relaxed diet groups. This would also avoid significant age differences between relaxed and strict diet groups. Age can be a confounder in terms of protein tolerance [14] and compliance.

In conclusion, this study shows non-inferiority of a simplified diet with 3 categories compared to a strict diet in terms of tyrosine levels. We advocate that neither weighing of foods nor calculation of protein or tyrosine intake is necessary. Phenylalanine levels were even higher which may have a positive impact on neurocognitive development. Simplification of dietary recommendations may improve the compliance and quality of life of patients and their families. Nonetheless, relaxation of the diet should be regularly monitored in order to assess metabolic control and not to risk organ dysfunction as well as malignancy.

## Figures and Tables

**Figure 1 nutrients-13-00134-f001:**
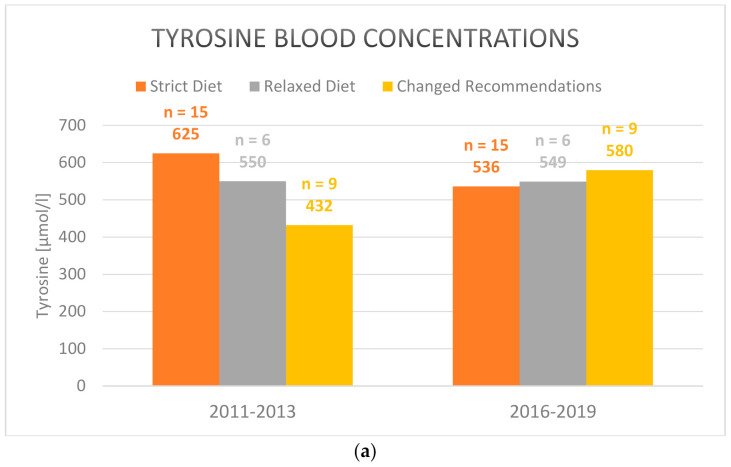
Mean (**a**) tyrosine and (**b**) phenylalanine blood concentrations in plasma under strict, relaxed/simplified and changed diet in the 2011–2013 and 2016–2019 periods (2016–2019 vs. 2011–2013 for the group with changed recommendations after the workhsop: tyrosine blood concentrations: *p* = 0.007; phenylalanine blood concentrations: *p* = 0.153). Only those patients are included who participated both in the pre- (2011–2013) and post-workshop surveys (2016–2019).

**Table 1 nutrients-13-00134-t001:** Simplified choice of food by using three categories according to protein content as developed in the workshop in 2013.

Unrestricted (Low Protein)	Occasional Consumption Possible (Moderate Protein)	Inappropriate (Protein Rich)
Fruits, vegetables, potatoes, rice, butter, oil, cream, crème fraiche, sour cream, sugar, sweets, rice/oat drink, special low-protein products	High-fat dairy products (e.g., cream cheese), normal bread, pasta, pastries, nuts, legumes, eggs processed in food	Meat, sausages, eggs, poultry, fish, sea food

**Table 2 nutrients-13-00134-t002:** Dietary recommendations at single centers: Calculation of protein and tyrosine in meals and diet regarding strict or simplified food selection are shown for the 2010–2013 and 2016–2019 periods. The target values for tyrosine and phenylalanine blood concentrations are listed.

Center	Protein/Tyrosine Calculation before 2013	Protein/Tyrosine Calculation after 2013	Diet before 2013	Diet after 2013	Tyr Target values	Phe Target Values
1	both	both	strict	strict	<500	normal
2	both	both	strict	strict	<400	<40
3	no data ^1^	no calculation	no data ^1^	strict	<400	50
4	protein only	protein only	strict	strict	<500	normal
5	protein only	no calculation	strict	simplified	<400	>50
6	no calculation	no calculation	strict	simplified	<400	>30
7	tyrosine only	tyrosine only	simplified	simplified	<400	normal
8	tyrosine only	tyrosine only	simplified	simplified	<400	20–80
9	protein only	protein only	simplified	simplified	<800	30–80

^1^ Center 3 only started treating Tyr 1 patients after 2013. Patient data for the period 2011–2013 was collected with the help of another participating center which treated these patients up to 2013.

**Table 3 nutrients-13-00134-t003:** Mean tyrosine (Tyr) and phenylalanine (Phe) concentrations in µmol/L from the 2011–2013 (pre-workshop) and 2016–2019 periods (post-workshop).

Center	Number of Patients	Mean Age (std. dev.)	Tyr 2011–2013	Tyr 2016–2019	Phe 2011–2013	Phe 2016–2019	Diet after 2013
1	9	14.3 (+/−6.9)	645.6	599.4	58.9	54.7	strict
2	10	7.2 (+/−5.3)	555.6	447.2	42.9	35.5	strict
3	3	21.3 (+/−4.9)	680.5	640.2	42.0	45.5	strict
4	2	27.5 (+/−5)	522.8	722.1	57.7	46.1	strict
5	5	12.1 (+/−4.3)	426.7	649.1	61.9	111.2	simplified
6	6	13.8 (+/−6.6)	391.6	475.6	32.9	40.1	simplified
7	5	16.7 (+/−6.1)	523.9	542.9	60.7	64.4	simplified
8	2	7.5 (+/−0.7)	- ^1^	463.8	- ^1^	30.5	simplified
9	2	22 (+/−1.4)	601.4	647.5	44.0	51.8	simplified
total	44 ^2^	13.8 (+/−7.4)	552.1	551.0	51.1	54.5	

^1^ patients were treated by a different center before 2013 which did not participate in this study. ^2^ Two patients from center 5 and one patient from center 7 were not included in this calculation due to liver transplantation.

**Table 4 nutrients-13-00134-t004:** Mean tyrosine (Tyr) and phenylalanine (Phe) concentrations in µmol/L for centers with strict or simplified diet in the period 2016–2019; *p*-values: strict vs. simplified diet.

Dietary Recommendations	Number of Patients	Tyr 2016–2019	Phe 2016–2019
strict diet	24	551	44.8
simplified diet	20	552	65.5
*p*		0.995	0.028

## Data Availability

The data presented in this study are available on request from the corresponding author. The data are not publicly available due to pseudonomization of patients and the interest of single centers not to disclose the identity of their patients.

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
