# Peer review of "Hepatorenal Tyrosinaemia: Impact of a Simplified Diet on Metabolic Control and Clinical Outcome"

_nutrients, 2020, doi:10.3390/nu13010134_

Round 1

Reviewer 1 Report

Thank you for you paper.  A difficult topic due to the rareness of the condition but a very important paper for those with and whom care for this disease. Well written and presented.

Very mild editing

Page 2 line 52 by 'concentrations' do you mean 'intake'? -I presume you are talking about the food-the next sentence is about plasma so it is confusing

Page 8 line 267 what do you mean by 'dystrophy'

How come three of the patients had liver transplantation and are in the study? Are they on diet?? 

Author Response

Dear Reviewer,

First of all, we thank you for your helpful comments and suggestions.

As we were asked to shorten our manuscript considerably some of the issues you referred to were eliminated from the text.

We made the following changes in response to your comments:

Page 2 line 51 changed to:

“…led to increased tyrosine and phenylalanine blood concentrations without weighing food.”

Here we talk about the blood concentrations, not food intake. Blood concentrations are our main variable since it is more precise than food intake. Now I hope it is less confusing and clear that we talk about the plasma/blood concentrations in this and the following sentence.

Page 8 line 267: This part of the manuscript was deleted. We were referring to low bodyweight and height/growth in comparison to other children at the same age.

Patients with liver transplantation: Two centers included transplanted patients in the questionnaires. As these patients do not receive protein-reduced diet we now deleted their d

Reviewer 2 Report

The authors nicely show that multicenter studies, although demanding in terms of study design and patient/data compilation, are a valuable tool for reviewing current therapeutical standards to improve the balance between metabolic control and quality of life. 

It's nice to read that a workshop was basis for a multicenter study and that the results of the workshop were followed up. 

Minor changes to be considered:

Line 23: Affiliation No.8 not found in author listing (line 4 to 9)

Line 75: nitsinone to nitisinone

Line 162/163: eight, two and one center = 11 centers instead of 9? Compared to Table 2 it should be six centers <400

Line 255:  Mean Phe level 60 or 70 umol/l? Table 6 states 60 vs. Figure 1b stating 70

Line 315: Table 7 does not give that information, but Table 9

Line 330: Table 5 does not give that information, but Table 6

Figures in Appendix show invalid characters in title, double check correct display in final print approval

Author Response

Dear Reviewer,

First of all, we thank you for your helpful comments and suggestions.

Some of the references you made don’t exist anymore, because one main critic of the editor was the paper’s length.

Line 23: there is a double affiliation for the author Michel Hochuli. This has now been corrected.

Line 74: nitsinone was changed to nitisinone

Line 162/163, now 151 f.: there was an error in the written text concerning the number of centers. Table 2 is correct. The text has now been corrected:

Six of the centers recommended tyrosine levels of <400, two <500 µmol/l and one center aimed for values <800 µmol/l”

Table 6 vs Fig. 1b: The Phenylalanine concentrations in the figure are correct, 70 instead of 60 µmol/l. Table 6 was deleted to shorten the paper according to suggestions by the editor.

Former line 315: You are right, the automatic numbering apparently didn’t work in this case. It should be a reference to table 9, not table 7. However, this part of the paper was deleted.

Lines 211, 213, 220: These lines refer to table 4 (former table 5). As the table refers to the information in the text, we didn’t change the numbering. Former table 6 was deleted.

We checked the appendix and changed the formatting so the final print should be fine.
